# A visual inductive priors framework for data-efficient image classification.

**Abstract.** State-of-the-art classifiers rely heavily on large-scale datasets, such as ImageNet[7], JFT-300M[22], MSCOCO[18], Open Images[16], etc. Besides, the performance may decrease significantly because of insufficient learning on a handful of samples. We present Visual Inductive Priors Framework (VIPF), a framework that can learn classifiers from scratch. VIPF can maximize the effectiveness of limited data. In this work, we propose a novel neural network architecture: DSK-net, which is very effective in training from small data sets. With more discriminative feature extracted from DSK-net, overfitting of network is alleviated. Furthermore, A loss function based on positive class as well as a induced hierarchy are also applied to further improve the VIFP's capability of learning from scrach. Finally, we won the **1st Place** in VIPriors image classification competition.

**Keywords:** learn from scratch, classification, visual inductive priors, DSK-net, induced hierarchy

## 1 Introduction

Convolutional Neural Networks (CNNs) have achieved state-of-the-art performance in image classification, object detection, semantic segmentation, etc. With the appearance of AlexNet[15], VGG[21], Inception[24, 13, 25, 23], ResNet[10], EfficientNet[26], ResNeSt[33], etc., the top-1 accuracy on ImageNet has been increased from 62.5%(AlexNet) to 84.5%(ResNeSt-269). Besides different network backbones, there are also many plug-and-play modules which can significantly improve accuracy, such as SE(Squeeze-and-Excitation)[12], CBAM(Convolutional Block Attention Module)[30], ECA (Efficient Channel Attention)[28], etc.

However, due to the limitation of label data, the performance of CNN is greatly limited. Pre-trained models are the most common solution that can get a fine result because of the prior knowledge. But there are only few pre-trained models which are fixed architectures and proposed. For training from scratch on VIPriors classification dataset which has only 50 training samples per class, the effectiveness of learning plays an important role. Effective and sufficient augment strategies are necessary, such as rand erasing[36], Mixup[34], CutMix[32], Cutout[8], AutoAugment[5], RandAugment[6], etc. On the other hand, models would overfit easily with little training data, so it is crucial to lighten the overfitting with appropriate regularization.

In this work, a novel network architecture Dual Selective Kernel network(DSK-net) is proposed to improve the effectiveness on small scale datasets. For more data-efficient learning, positive class classification loss and intra-class compactness loss are applied to enhance discriminative power of the deeply learned features. An induced hierarchy is used which is easier for models to learn from scratch. Methods are evaluated on VIPriors Image Classification dataset. The dataset is derived from ImageNet and contains 50 images per class for training and testing. Experimental results show that our methods achieve the best performance on VIPriors classification dataset.

## 2    Related works

### 2.1    Data Augmentation

Augmentation is an effective way to improve CNNs' performance especially in the case of insufficient data. Mixup[34] trains a model on convex combinations of pairs of examples and their labels together. Cutout[8] randomly erases square regions on input images during training. CutMix[32] cuts and pastes patches among training images where the training labels are also mixed proportionally to the area of patches. It can efficiently make use of training pixels and retain the regularization effect of regional dropout. GridMask[3] drops pixels on the input images with multiple squares and different ratios. Recently, with the emergence of AutoML, network learning strategies also can be searched from data. Auto-Augmentation[5] is a series of augmentation operation strategies searched on ImageNet which needs a huge space for searching. Hence RandAugmentation[6] proposes a simplified search space which has less computational expense.

### 2.2    Translation invariance in CNNs

It is generally known that CNNs are not shift-invariant. A small shift or translation of input will result in a quite different output. To reduce the influence of translation, several augmentation operations are often used such as scaling, rotation and reflection[2][4][9][20][31]. [35] integrates low-pass filtering to anti-alias which is a common signal processing module. [14] proposes a full convolution architecture by removing spatial location as feature which improves equivariance and invariance of the inductive convolutional prior.

### 2.3    Important feature learning

For image classification, locating and recognizing the discriminative feature is the key to a better performance. And most of discriminative feature extraction modules are based on attention mechanisms which is inspired by human brain neural units. SE(Squeeze-and-Excitation)[12] and ECA[28] are channel attention architecture. Channel and spatial attention modules are applied in CBAM[30]. Inspired by adaptive field sizes of neurons, [17] proposes Selective Kernel(SK)

convolution which is based on soft-attention manner to improve feature extraction efficiency. Except attention architecture, loss function can also help model learn more discriminative feature. Center loss[29] is implemented by increasing inter-class dispersion and intra-class compactness. It learns centers form deep features of each class, and then penalizes the distances between deep features and their corresponding class centers.

## 3 Proposed Method

To be more data-efficient, firstly, a 3-branched network called Dual Selective Kernel(DSK) network is proposed in Fig.1. DSK has the advantages of discriminative feature extraction, translation invariant and regularization. Secondly, a composite loss function is designed to improve feature discrimination. It helps models not only classify correctly but also increase the diversity of different classes.

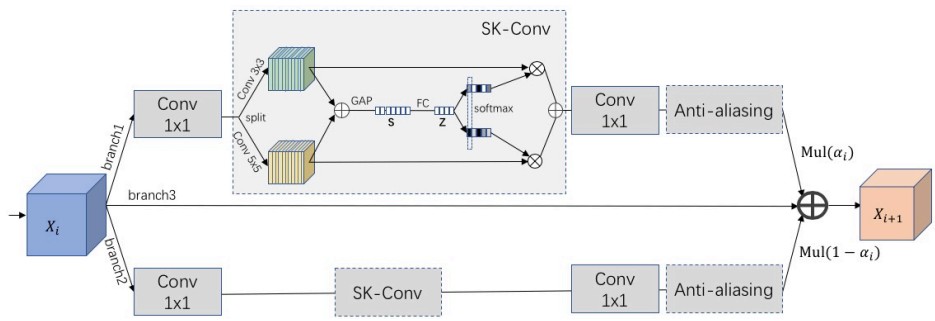

**Fig. 1.** Dual selective kernel residual block.

### 3.1 Dual selective kernel network

**Discriminative feature extraction** To adjust the receptive fields of neurons automaticly, selective kernel convolution[17] is added into residual block. For any given feature map $X \in \mathbb{R}^{H \times W \times C}$, $X$ is respectively conducted by convolutions of kernel size 3 and 5. Then we two transforms are conducted: $\widehat{\mathcal{F}}: X \to \widehat{\mathcal{U}} \in \mathbb{R}^{H \times W \times C}$ and $\widetilde{\mathcal{F}}: X \to \widetilde{\mathcal{U}} \in \mathbb{R}^{H \times W \times C}$. Both $\widehat{\mathcal{F}}$ and $\widetilde{\mathcal{F}}$ are composed with depthwise convolution, Batch Normalization and ReLU. Feature $\mathcal{U}$ is a element-wise sum of $\widehat{\mathcal{U}}$ and $\widehat{\mathcal{U}}$. For $\mathcal{U}$, global average pooling is used for information embedding. Further, a compact feature $s \in \mathbb{R}^C$ is created by passing feature embedding to fully connected layer for squeeze. Then Batch Normalization, ReLU and another two fully connected layers are applied for different

kernel excitation. Finally, a soft attention across channels is conducted to select information in different spatial scales. The weights $\widehat{\omega}$ and $\widetilde{\omega}$ for attention is calculated by a channel-wise softmax operation. The final feature map is obtained by applying attention weights to feature $\widehat{\mathcal{U}}$ and $\widehat{\mathcal{U}}$:

$$\mathcal{V} = \widehat{\omega} \cdot \widehat{\mathcal{U}} + \widetilde{\omega} \cdot \widetilde{\mathcal{U}} \tag{1}$$

**Translation invariant** The reducing spatial resolution operations in CNNs including max pooling, average pooling and strided convolution are harmful to shift-equivariance. Blur pool[35] is an anti-aliased architecture which is compatible with above architectures components. For example, max pooling with stride=2 in CNNs will be split into max pooling with stride=1 and blur pool with stride=2. And strided convolution with activation function will be split into convolution with stride=1, activation function and blur pool. As for blur pool kernel, it has several anti-aliasing filters from size 2x2 to 5x5 with increasing smoothing. In DSK, 3x3 filter is applied in max pooling and strided convolution.

**Regularization** Like data augmentation techniques applied to input data, it is reasonable to apply corresponding techniques to representation branch in residual block. Let $X_i$ denotes the input tensor of residual block $i$. $\mathcal{W}_i^1$ and $\mathcal{W}_i^2$ denote weights associated with the two residual units. $\mathcal{F}$ denotes the residual function and $X_{i+1}$ denotes the outputs from $i$. The 3-branch architecture can be represented as:

$$X_{i+1} = X + \lambda_i \mathcal{F}(X_i, \mathcal{W}_1) + (1 - \lambda_i)\mathcal{F}(X_i, \mathcal{W}_2) \tag{2}$$

When forward and backward during training, $\lambda_i$ is a random value of 0 or 1. And $\lambda_i$ is 0.5 for inference.

### 3.2   Loss function

Categorical cross-entropy(CE) loss after softmax is widely used in multi-class classification. But for VIPriors classification dataset, CE is suboptimal. Because it forces models to only focus on training image and ignore the compactness of intra-class. In this section, several loss functions will be discussed and a combined loss is proposed as Eq.3 for a better performance.

$$L = \alpha L_{PCL} + \beta L_{CL} + \gamma L_{TSL} \tag{3}$$

**Positive class loss** CE loss is showed in Eq.4. Let $p$ represents the output of a model and $l$ represents one-hot labels. CE not only directs model to classify the ground truth class correctly but also forces the prediction of other classes as low as possilble.

$$L_{CE} = -\frac{1}{N}\sum(l * log(p) + (1 - l) * log(1 - p)) \tag{4}$$

But is it suitable to use loss on a small dataset in which the number of classes is far greater than the number of samples per class? Additionally, [19] proves that there are many label errors in ImageNet including actual multi-label images but only labeled with singe class label. We have reasons to believe that there is the same question on VIProirs classification dataset. Based on the above, making models only focus on ground truth label may be more beneficial during learning. Consequently, the positive class loss(PCL) is proposed as:

$$L_{PCL} = \frac{1}{N} \sum (-l * log(p) + (1 - cos(l, p)))$$  (5)

PCL has two parts: the former is from CE, the latter is cosine loss[1].

**Center loss** Although PCL can direct model for a better learning, it is easily overfitting with less data. Therefore, center loss(CL)[29] in Eq.6 is used for more discriminative feature extraction. Let $x_i \in \mathbb{R}_d$ denote the $i$th deep feature belonging to the $y_i$th class. The $y_i$th class center of deep features $c_{y_i} \in \mathbb{R}_d$ is computed by averaging $y_i$th class features of the corresponding classes in each iteration.

$$L_{CL} = \frac{1}{2} \sum_i ||x_i - c_{y_i}||_2^2$$  (6)

**Tree supervision loss** The semantic relations of classes in VIPriors can be induced as a hierarchical tree. Child nodes of the tree represent 1000 classes in the dataset and parent nodes represent superclasses such as animal, vehicle and etc. For every parent node, its child nodes often have some commonalities which is helpful for classification. Inspired by Neural-Backed Decision Trees(NBDT)[27], a hierarchical architecture is defined according to the semantic relationship based on 1000 classes. Tree supervision loss(TSL) is used for model training. Let $x \in \mathbb{R}_d$ denotes featurized sample, $w_{i,j}$ denotes the edge from $i$-th leaf nodes to root node on $j$-th layer. TSL can be represented as:

$$L_{TSL} = L_{CE}(\sum_i \sum_j \langle w_{i,j}, x \rangle, l)$$  (7)

## 4   Experiments

### 4.1   Implementation Details

To efficiently use the training pixels and retain the regularization effect of region dropout, following data augmentation methods are used in our models: random resize and crop, random horizontal flip and CutMix(with a probability of 0.5). And all models are trained with 16 GPUs and 64 samples per CPU. In the training stage, warm up with initial lr of 0.0001 in 5 epochs, cosine learning rate[11] with initial lr of 0.1, dropout with probability of 0.2, weight decay of 0.0001 and label smooth are used for learning. For coefficients in Eq.6, $\alpha, \gamma$

and $\beta$ are set to 1, 0.0005 and 1. In early time of the competition, we trained model on training set for methods attempt and verificaiton. And in the final stage, we trained models on both training set and most of validation set. Only a little samples in validation set were reserved for validation. For final prediction, Test Time Augmentation(TTA) with 10-crop was used. Additionally, sufficient training epochs at least 360 can also improve model performance. Experimental results prove that increasing training epochs from 90 to 360 improve model accuracy by 5.3%.

## 4.2   Results

Table 1 shows the results for ResneXt(2-branch architecture), D-ResNeXt(3-branch architecture), SK-ResNeXt(2-branch architecture), DSK-ResNeXt(3-branch architecture), PSL and CL on validation set. Models are trained with 360 epochs.

**Table 1.** Performance of DSK-net, PSL and CL on validation set.

|                                            | top-1 acc. (%) |
| ------------------------------------------ | -------------- |
| ResNeXt50_32x4d                            | 52.01          |
| D-ResNeXt50_32x4d                          | 54.45          |
| SK-ResNeXt50_32x4d without anti-aliasing   | 54.06          |
| SK-ResNeXt50_32x4d                         | 54.37          |
| DSK-ResNeXt50_32x4d                        | 55.97          |
| DSK-ResNeXt50_32x4d+PSL                    | 56.48          |
| DSK-ResNeXt50_32x4d+PSL+CL                 | **57.51**      |

Table 2 shows results of TSL for EfficientNet and ResNeSt in the final stage. Models are trained with 720 epochs and tested on partial validation set.

**Table 2.** The experiment results of TSL.

|                              | top-1 acc. (%) |
| ---------------------------- | -------------- |
| EfficientNet-b3              | 62.42          |
| EfficientNet-b3+TSL          | 63.15          |
| EfficientNet-b5              | 65.43          |
| EfficientNet-b5+TSL          | 65.85          |
| EfficientNet-b6              | 65.67          |
| EfficientNet-b6+TSL          | 66.26          |
| ResNeSt-101(320x320)         | 65.96          |
| ResNeSt-101(320x320)+TSL     | 67.15          |
| ResNeSt-200(320x320)         | 67.40          |
| ResNeSt-200(320x320)+TSL     | **67.81**      |

Table 3 shows the results of DSK-net in the final stage. Models are trained with 540 epochs and tested on partial validation set. 69.59% of DSK-ResNeXt101_32x4d is the best single model performance we achieved.

**Table 3.** The experiment results of DSK-net.

|  | top-1 acc. (%) |
|---|---|
| DSK-ResNeXt50_32x4d(224x224) | 67.35 |
| DSK-ResNeXt50_32x4d(320x320) | 69.20 |
| DSK-ResNeXt101_32x4d(224x224) | 68.02 |
| DSK-ResNeXt101_32x4d(320x320) | **69.59** |

### 4.3   Other tricks

Results for CutMix showed in Table 4 indicate that the global semantic information and local area feature are equally import.

**Table 4.** The experiment results on validation set for CutMix. Input size is 320x320, training epoch is 90.

|  | top-1 acc. (%) |
|---|---|
| ResNeXt50_32x4d | 45.23 |
| ResNeXt50_32x4d+CutMix with prob=0.3 | 45.73 |
| ResNeXt50_32x4d+CutMix with prob=0.5 | **46.25** |
| ResNeXt50_32x4d+CutMix with prob=0.7 | 45.85 |
| ResNeXt50_32x4d+CutMix with prob=1.0 | 44.35 |

Results of label smooth, dropout and dual pool are showed in Table 5:

**Table 5.** The experiment results of label smooth, dropout and dual pool on validation set. Models are trained with 360 epochs.

|  | top-1 acc. (%) |
|---|---|
| ResNeXt50_32x4d | 50.56 |
| ResNeXt50_32x4d+dual pool | 50.87 |
| ResNeXt50_32x4d+label smooth | 50.70 |
| ResNeXt50_32x4d+dropout with prob=0.2 | 50.90 |
| ResNeXt50_32x4d+dropout with prob=0.4 | 50.84 |
| ResNeXt50_32x4d+dual pool+label smooth+dropout with prob=0.2 | **51.82** |

### 4.4   Ensembling

For a better performance, we ensembled predictions of above methods in total 16 models including EfficientNet-b5, EfficientNet-b6, ResNeSt-101, ResNest-200, DSK-ResNeXt50, DSK-ResNeXt101. Finally, a weighted score average method was used that the weight of higher performance models was 3, the rest was 1. Finally, we got the score of 73.08% on test set.

Fig.2 shows an overview of methods and appearances. No external image/video data or pre-trained models were used throughout the competition.

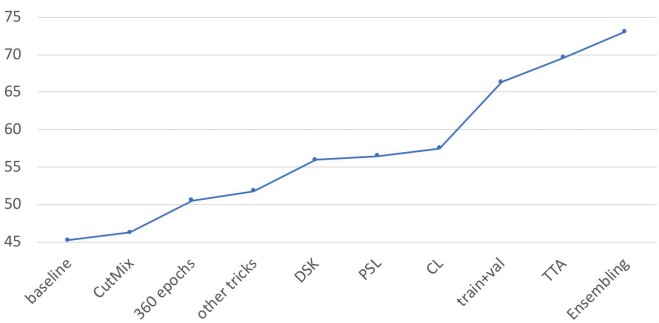

**Fig. 2.** Perfromance overview.

## 5   Conclusions

In this paper, we discuss and explore data-efficient learning, visual inductive priors and training from scratch. In VIPF, we propose a novel architecture called DSK-net, which is robust to translation invariance. Sufficient experiment results fully proved that DSK-net learns efficiently from insufficient data and outperformed EfficientNet, ResNeSt on VIPriors classification dataset. Then a loss based on positive class is applied for model constraint. And an induced hierarchy is used which can direct models to learn discriminatively and easily. Experimential results show that VIPF we proposed is effective. Finally we won the 1st place in VIPriors image classification competition.

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
