# OpenReview forum: "A visual inductive priors framework for data-efficient image classification."
_thecvf.com/ECCV/2020/Workshop/VIPriors — VIPriors Poster_

### Official Review · AnonReviewer2 · 2020-07-22
**Interesting method, unclear explanations**

**Confidence:** 4
**Rating:** 6

**Review:**

#### 1. [Summary] In 2-3 sentences, describe the key ideas, experiments, and their significance.
The paper proposes the Dual Selective Kernal residual building block consisting of Selective Kernel convolutions [17] and BlurPool layers [35]. Furthermore, a novel positive class loss is introduced motivated by the low number of samples per class and possible label errors, and a tree supervision loss to incorporate semantic relationships amongst the ImageNet classes. Different models are trained and evaluated on the VIPriors classification dataset where significant performance improvements are shown.

#### 2. [Strengths] What are the strengths of the paper? Clearly explain why these aspects of the paper are valuable.
* The method seems effective
* Ablation studies have been performed to show the effectiveness of the individual contributions (i.e. DSK and PSL)
* Apart from several unclarities in the method section (see 3.), the paper is easy to read.

#### 3. [Weaknesses] What are the weaknesses of the paper? Clearly explain why these aspects of the paper are weak.
* The method section contains some unclarities:
 * (lines 128-129) What are the the two transforms $\mathcal{\hat{F}}$ and $\mathcal{\tilde{F}}$ and where are these in Figure 1? Do you first perform regular 3x3 and 5x5 convolutions, followed by depthwise convolutions? I also don't see the Batch Normalization and ReLU layers and $\mathcal{U}$ in Figure 1.
 * (line 132) "a compact feature $s$..."
  $s$ should be $z$ according to Figure 1?
 * (line 135) "soft attention across channels is conducted"
  It is unclear how the softmax operation is applied. Is it applied per channel between $\hat{\omega}$ and $\tilde{\omega}$ or across all channels of $\hat{\omega}$ and all channels of $\tilde{\omega}$ separately? The former seems more reasonable but the text is not very explicit about it.
 * $\lambda$ in equation (2) and lines (160-162) is $\alpha$ in Figure 1?
 * lines (168-169) "compactness of intra-class" sounds a bit ambiguous. Could you elaborate on this?
 * Can you explain the positive class loss as introduced in equation (5)? What is the motivation behind chosing the cosine loss? Please elaborate.
 * What is $l$ in equation (7)? Where do you extract $w_{i,j}$ and $x$ from? Can you intuitively explain how the loss function works and what is optimized?
 * What are the motivations for the model choices? It seems a bit arbitrary that ResNeXt is combined with PSL and CL but not with TSL, and ResNeSt is combined with TSL but not with PSL and CL.
* The method is only evaluated on a single dataset. It would have been nice if the authors had shown that their method also generalizes to other settings.

#### 4. [Overall rating] Paper rating
6. Marginally above acceptence threshold.

#### 5. [Justification of rating] Please explain how the strengths and weaknesses aforementioned were weighed in for the rating.
The methods introduced by the authors seem effective and therefore I am willing to accept the paper, in the hope that the authors will clarify and extend the method section for the camera-ready version.

#### 6. [Detailed comments] Additional comments regarding the paper (e.g. typos or other possible improvements you would like to see for the camera-ready version of the paper, if any.)
* See unclarities in 3.
* Please combine multiple references in a single bracket, i.e. [1][2][3] should become [1,2,3] and use ~\cite{} instead of \cite{} for extra spacing.
* Small typos:
 * line 18: "VIFP" --> VIPF
 * line 88: architecture --> architectures
 * line 128: "Then we two transforms" --> remove "we"
 * line 131: "element-wise sum of u-hat and u-hat" --> one of them should be tilde
 * line 138: same as above: one of u-hat should be u-tilde
 * etc. Please check your writing for the camera-ready version.

---

### Official Review · AnonReviewer1 · 2020-07-27
**A visual inductive priors framework for data-efficient image classification**

**Confidence:** 5
**Rating:** 7

**Review:**

#### 1. [Summary] In 2-3 sentences, describe the key ideas, experiments, and their significance.
The paper smartly combines different loss functions and augmentation techniques to overcome the problem of small classification dataset. In addition, it proposes new dual architecture by using Selective Kernel convolution.

#### 2. [Strengths] What are the strengths of the paper? Clearly explain why these aspects of the paper are valuable.
- Proposed model
- Performance

#### 3. [Weaknesses] What are the weaknesses of the paper? Clearly explain why these aspects of the paper are weak.
- Clarity (explanation of models, experiments and results)


#### 4. [Overall rating] Paper rating
7

#### 5. [Justification of rating] Please explain how the strengths and weaknesses aforementioned were weighed in for the rating.


#### 6. [Detailed comments] Additional comments regarding the paper (e.g. typos or other possible improvements you would like to see for the camera-ready version of the paper, if any.)
- Please explain why each related work part is related to the paper.
- L.115: (Fig.1) The notations in the figure are not same in the equations. Fig.1 is not refered in the text.
- Please explain each notation in each equation.
- Please explain each proposed method.
- L.161: How do you choose λi? Is it dropped from one side in each time? Is there any cases that both SK-branches can be kept or dropped?
- L.344:In the conclusion part, it is said that DSK-net is robust to translations. Is there any experiments to show that?
Missing citations:
- L.28: "image classification[???], object detection[???], semantic segmentation[???]"
- L.37: what are those "few pre-trained models"?
Typos:
- L.15: effiective -> effective or efficient
- L.17: "A" loss (lowercase)
- L.147, 221, 347: Starting the sentence with "And" (problem)
- L.176: possible
- L.180: a loss
- L.271: the word is not fitting into line.
- L.344: robust to translations.

---

### Decision · Program_Chairs · 2020-07-29

**Decision:**

Accept (Poster)

**Comment:**

It is our pleasure to inform you that your paper has been accepted to the poster track of 1st Visual Inductive Priors for Data-Efficient Deep Learning Workshop.

Please note the following deadlines:
* August 11, 2020 - workshop material, including:
 * paper in PDF format;
 * pre-recorded video presentation;
 * slides of the presentation in PDF.
* September 15, 2020 - camera-ready paper

The reviews can be found on OpenReview. Please take these comments and suggestions into account when preparing the camera-ready version of your paper, which is due September 15, 2020. The camera-ready paper should be uploaded to OpenReview.

As part of the workshop, each accepted paper must submit a pre-recorded 90 second talk before August 11, 2020. You will receive more information on how to upload the material shortly. The requirements for the video are:
* Duration: maximum 90 seconds
* MP4 format
* File size max. 100 MB
* Has an inset with a video of the speaker
* 16:9 aspect ratio (strongly preferred)
* 1920x1080 resolution (strongly preferred, at least 720 height)

Our suggested software for pre-recording your presentation is Zoom. For more information, please refer to the following guides:
How to record with Zoom Guide: http://homepages.inf.ed.ac.uk/rbf/ECCV2020HowtoRecordusingZoom.pdf
How to Record with Zoom tutorial: https://www.youtube.com/watch?v=CR199W7HdC0
Please ensure that at least one of the authors of the paper is available to attend the workshop during the allotted times. Note that the workshop will take place in two sessions spread across time zones (details are to follow). We will send instructions on how to connect to the workshop as soon as possible. The schedule for all talks and papers will be posted soon at the workshop website: https://vipriors.github.io.

We look forward to seeing you at the workshop!